# Domain Adaptation for Medical Image Segmentation: A Meta-Learning Method

**DOI:** 10.3390/jimaging7020031

**Published:** 2021-02-10

**Authors:** Penghao Zhang, Jiayue Li, Yining Wang, Judong Pan

**Affiliations:** 1School of Information and Communication Engineering, Beijing University of Posts and Telecommunications, Beijing 100876, China; zphbupt@bupt.edu.cn; 2School of Computing, Informatics, and Decision Systems Engineering, Arizona State University, Tempe, AZ 85281, USA; 3Department of Radiology, State Key Laboratory of Complex Severe and Rare Diseases, Peking Union Medical College Hospital, Chinese Academy of Medical Sciences and Peking Union Medical College, Beijing 100730, China; WangYiNing@pumch.cn; 4Department of Radiology and Biomedical Imaging, University of California San Francisco, San Francisco, CA 94143, USA; judong.pan@ucsf.edu

**Keywords:** medical image segmentation, domain adaptation, meta-learning, U-Net

## Abstract

Convolutional neural networks (CNNs) have demonstrated great achievement in increasing the accuracy and stability of medical image segmentation. However, existing CNNs are limited by the problem of dependency on the availability of training data owing to high manual annotation costs and privacy issues. To counter this limitation, domain adaptation (DA) and few-shot learning have been extensively studied. Inspired by these two categories of approaches, we propose an optimization-based meta-learning method for segmentation tasks. Even though existing meta-learning methods use prior knowledge to choose parameters that generalize well from few examples, these methods limit the diversity of the task distribution that they can learn from in medical image segmentation. In this paper, we propose a meta-learning algorithm to augment the existing algorithms with the capability to learn from diverse segmentation tasks across the entire task distribution. Specifically, our algorithm aims to learn from the diversity of image features which characterize a specific tissue type while showing diverse signal intensities. To demonstrate the effectiveness of the proposed algorithm, we conducted experiments using a diverse set of segmentation tasks from the Medical Segmentation Decathlon and two meta-learning benchmarks: model-agnostic meta-learning (MAML) and Reptile. U-Net and Dice similarity coefficient (DSC) were selected as the baseline model and the main performance metric, respectively. The experimental results show that our algorithm maximally surpasses MAML and Reptile by 2% and 2.4% respectively, in terms of the DSC. By showing a consistent improvement in subjective measures, we can also infer that our algorithm can produce a better generalization of a target task that has few examples.

## 1. Introduction

Image segmentation is often the first and the most critical step in the analysis of medical images for computer-aided diagnosis and therapy. Medical image segmentation is a challenging and complex task due to the intrinsic nature of images. For instance, it is difficult for experienced experts to accurately identify multiple sclerosis lesions in MRIs due to the variability in lesion location, size, and shape, and the anatomical variability across patients [1]. Manual segmentation has been gradually replaced by automatic segmentation because of the high costs and time consumption [2]. Among existing automatic segmentation methods, Convolutional neural networks (CNNs) have demonstrated great achievement in increasing segmentation accuracy and stability [3,4,5,6,7,8]. However, existing CNNs are limited by the problem of dependency on the availability of training data owing to high manual annotation costs and privacy issues.

To counter that limitation, domain adaptation (DA) and few-shot learning have been extensively studied in semantic segmentation. DA focuses on using some data from the target domain to quickly adapt a model trained in the source domain [9,10,11,12,13,14]. Few-shot learning aims to learn the patterns of new concepts unseen in the training data, given only a few labeled examples [15,16,17,18,19]. Inspired by previous works on DA and few-shot learning, we have a question: can we adjust an optimization algorithm so that the segmentation model can be good at learning with only a few examples? In order to solve this question, we take an optimization-based meta-learning method to DA. The meta-learning method learns to align source and target data in a domain-invariant discriminative feature space [20,21,22]. Existing optimization-based meta-learning algorithms such as model-agnostic meta-learning (MAML) [23] and Reptile [24] aim to search for the optimal initialization state to quickly adapt a base-learner to a new task. MAML is compatible with any model trained with gradient descent. It is also applicable to a variety of different learning problems, including classification, regression, and reinforcement learning. MAML provides a good initialization of model parameters which achieve optimal fast learning toward a new task with only a small number of gradient steps. In the meantime, MAML can avoid the overfitting that may happen when using a small dataset. In MAML, source tasks are split into support and query sets for support-training and query-testing purposes, respectively. In the inner loop of MAML, a model is trained to solve each support set in turn based on a few examples and gradient steps. Fast domain adaptation is achieved by training the source model with the query set in the outer loop. Unlike MAML, which uses a support-query scheme to quickly adapt a model to a new task, in the inner loop of Reptile, the model is iteratively trained on a sampled task by multiple gradient steps. The model is then updated towards the gradients learned from the sampled task in the outer loop. Reptile does not require differentiating through the optimization process, making it more suitable for optimization problems where many update steps are required.

Even though MAML and Reptile use prior knowledge to choose parameters that generalize well from few examples, both algorithms limit the diversity of the task distribution that they can learn from in medical image segmentation. Specifically, MAML updates the initial parameter vector towards the direction of a query-testing phase, which limits the capability of updating the initial parameter vector by learning from the tasks in the support-training phase. Reptile limits the ability of updating the model parameters since it does not learn from diverse tasks in the inner loop. In order to counter the limitations in MAML and Reptile, we propose to augment both algorithms with the capability to learn from diverse segmentation tasks across the entire task distribution. Specifically, our algorithm aims to learn from the diversity of image features which characterize a specific tissue type while showing diverse signal intensities. The reason that the proposed idea can benefit from signal intensities is described as follows. In MRI, the terms low, intermediate, and high signal intensities are used. Depending on the scan protocol, a tissue type is imaged as white if it has high signal intensities, as gray if it has intermediate signal intensities, and as dark gray/black if it has low signal intensities. We focus on a class of tissue types which move often: the heart is moving as it beats, the colon is moving as it digests, etc. Due to the movement, these tissue types show diverse image features regarding location, size, shape, and impact on the surrounding area. The image features are described by diverse signal intensities, such as high, intermediate, and low. The learning capability is therefore enhanced if we can learn from the diversity of image features which characterize a specific tissue type while showing diverse signal intensities. Figure 1 displays two example tissue types; each one shows diverse image features regarding location, size, shape, and impact on the surrounding area.

Our algorithm is briefly introduced as follows: In the inner loop, we first iteratively train the initial parameter vector on a batch of support sets via multiple gradient steps. Based on the parameter vector learned from the support batch, we then adapt the parameter vector to a batch of query sets via one gradient step in the outer loop. After that, we update the model towards the parameter vector learned from the query batch. Both support and query batches are sampled from the entire task distribution. To demonstrate the effectiveness of our method, we conducted experiments using a diverse set of segmentation tasks from the Medical Segmentation Decathlon and two meta-learning benchmarks: MAML and Reptile. U-Net [25] and Dice similarity coefficient (DSC) were selected as the baseline model and the main performance metric, respectively. The experimental result shows that our algorithm maximally surpasses MAML and Reptile by 2% and 2.4% respectively, in terms of the DSC. By showing a consistent improvement in subjective measures, we can also infer that our algorithm can produce a better generalization of a target task that has few examples. The contributions of our algorithm focus on two points:Unlike existing meta-learning algorithms which limit the capability of learning from diverse task distributions, we studied the feasibility of learning from the diversity of image features which characterizes a specific tissue type while showing diverse signal intensities.We propose an algorithm which can nicely learn from diverse segmentation tasks across the entire task distribution. The effectiveness of our algorithm is illustrated by showing consistent improvements in DSC and subjective measures.

## 2. Related Work

### 2.1. Convolutional Neural Networks

The concept of deep learning originates from the research of hierarchical artificial neural networks. Unlike traditional segmentation methods that only utilize low-level information such as pixel color, brightness, and texture, deep learning methods perform better on extracting semantic information. One of the deep learning methods is CNN. CNN is a kind of neural network with a special connective structure in hidden layers. With its rich feature extractors, some classic models such as AlexNet [26], VGG [27], GoogleNet [28], and ResNet [29] have been widely used in most computer vision tasks.

In the field of medical image processing, the fully convolutional network (FCN) [30] and U-Net [25] are commonly used. Since FCN is a pixel-wise classification model, it does not perform as well as U-Net for exploiting the relationship between pixels and boundary information of the up-sampling results. U-Net consists of a contraction path (encoder) and a symmetrical extension path (decoder) connected by a bottleneck. The encoder gradually reduces the spatial size of feature maps, which captures the context information and transmits it to the decoder. The decoder recovers the image details and spatial dimensions of the object through up-sampling and skip connections. Even though there has been a collection of variations of U-Net produced to improve segmentation accuracy, it still appears to be inadequate for a segmentation task which needs to learn from a limited amount of training data. Considering its satisfactory performance in medical image segmentation, we selected U-Net as the baseline model and applied the proposed meta-learning method to it.

### 2.2. Optimization-Based Meta-Learning Methods

The inspiration of meta-learning comes from the human learning process, which can adapt new tasks quickly according to a few examples [31]. The proposed meta-learning method in this paper is optimization-based. This category of methods like MAML [23] and Reptile are closely related to our method.

MAML aims to learn from a number of tasks T sampled from a distribution p(T). These tasks are composed of a support set $\tau^{s}$ and a query set $\tau^{q}$. MAML requires that $\tau^{s}$ and $\tau^{q}$ do not have any overlapping class. The algorithm attempts to find a desirable parameter vector θ for a given model. In each inner loop of MAML, as shown in Figure 2a, the model is learned from a support batch $\tau^{s}$ sampled from $T^{s}$ with a loss function $L_{\tau^{s}}\left(\theta ,\tau^{s}\right)$. A transitional parameter vector $\theta^{s}$ is obtained by updating θ through a number of gradient steps. In the outer loop of MAML, a query batch $\tau^{q}$ sampled from $T^{q}$ is then used to update $\theta^{s}$ to $\theta^{q}$ based on a query loss $L_{\tau^{q}}\left(\theta^{s},\tau^{q}\right)$. After that, $\theta^{q}$ is applied to the update of θ. In Figure 2a, we use arrows to represent the direction of update. The arrow directed from $\theta^{s}$ to $\theta^{q}$ is parallel to the arrow directed from θ to $\theta^{*}$, where the entire updating process ends with $\theta^{*}$.

Instead of using a support-query scheme, as shown in Figure 2b, in each inner loop, Reptile updates θ by learning from the same batch τ through multiple gradient descent steps. The direction of update in the outer loop is determined by θ and $\theta^{\prime }$, where $\theta^{\prime }$ is the transitional parameter vector obtained from the inner loop. Reptile focuses on learning from the same batch and improving generalization with a particular number of gradient descent steps. The entire updating process ends with $\theta^{*}$.

Differently from MAML and Reptile, as shown in Figure 2c, in the inner loop of our algorithm, we first train the model parameters on a support batch $\tau^{s}$ with multiple gradient steps. In the outer loop, we then adapt the model parameters to a query batch $\tau^{q}$ by one gradient step. In Figure 2c, both $\tau^{s}$ and $\tau^{q}$ are sampled from the entire task distribution. Figure 3 displays how MAML and our algorithm select $\tau^{s}$ and $\tau^{q}$ in the meta-training phase. Each training task (1 or 2) mimics the few-shot scenario, which includes three classes with two support examples and one query example. Each example in either the support set or the query set is randomly selected from a set of training examples, which is displayed in different colors. Each example includes three classes: H, S, and P which represent the MRI images from THE heart, spleen, and prostate, respectively. We can observe from Figure 3a that the examples for support and query purposes are split into two partitions. in our algorithm, as shown in Figure 3b, the examples in each training task are selected from diverse examples across the entire example distribution. MAML updates the initial parameter vector towards the direction of query-testing phase, which limits the capability of updating the initial parameter vector by learning from the examples in the support-training phase. Unlike MAML, our algorithm avoids the limitation by removing the boundary between support and query examples. By doing so, the parameter vector is updated by learning from the diversity of support and query examples. The capability of learning is therefore enhanced. The diversity can be interpreted as: for any tissue type in a set of examples, this tissue type shows diverse image features over location, size, shape, and impact on the surrounding area (as shown in Figure 1). The direction of update in the outer loop is determined by θ and $\theta^{\prime }$. The entire updating process ends with $\theta^{*}$.

## 3. Methodology

In this study, we used U-Net [25] as the baseline model and learned the initialization of U-Net from multiple source tasks. Based on the parameter vector learned from the source tasks, we fine-tuned the model on the target task. We aimed to train a U-Net that can produce good generalization performance on the target task. In this section, we first present the general form of our algorithm. After that, we provide some theoretical analysis to better explain why the proposed algorithm works.

### 3.1. Meta-Learning Domain Adaptation

As shown in Algorithm 1, let θ denote the initial parameter vector, and we use a parametrized function $f_{\theta }$ to represent the baseline model. For any batch of tasks $\tau^{s}$ sampled from p(T), when $f_{\theta }$ adapts to $\tau^{s}$, the parameter vector θ becomes $\theta_{i}^{\prime }$. The updated parameter vector $\theta_{i}^{\prime }$ is computed using *i* gradient steps on $\tau^{s}$. Let $f_{\theta_{i}^{\prime }}$ denote the updated model; the update on the gradient at the *i*th step is described as
(1)$$\theta_{i}^{\prime }=\theta_{i-1}^{\prime }-\alpha \nabla_{\theta_{i-1}^{\prime }}L_{\tau^{s}}\left(f_{\theta_{i-1}^{\prime }}\right),$$
where α is a fixed hyper-parameter and represents the learning-rate in the inner loop. $L_{\tau^{s}}\left(f_{\theta_{i-1}^{\prime }}\right)$ represents the loss function of model $f_{\theta_{i-1}^{\prime }}$ on $\tau^{s}$. $\theta_{i}^{\prime }$ can be obtained by optimizing $f_{\theta_{i-1}^{\prime }}$ with respect to $\theta_{i-1}^{\prime }$ on the same batch of tasks sampled from p(T). The meta-objective can be described as
(2)$$\min_{\theta_{i-1}^{\prime }}L_{\tau^{s}}\left(f_{\theta_{i}^{\prime }}\right)=L_{\tau^{s}}\left(f_{\theta_{i-1}^{\prime }-\alpha \nabla_{\theta_{i-1}^{\prime }}L_{\tau^{s}}\left(f_{\theta_{i-1}^{\prime }}\right)}\right).$$

The optimization on the entire meta-learning process is performed over parameters θ. We can obtain $\theta^{\prime }$ by updating $\theta_{i}^{\prime }$ based on another batch of tasks $\tau^{q}$ which is also sampled from p(T). The optimization process is therefore described as
(3)$$\theta \leftarrow \theta +\beta [\theta_{i}^{\prime }-\alpha \nabla_{\theta_{i}^{\prime }}L_{\tau^{q}}\left(f_{\theta_{i}^{\prime }}\right)-\theta ],$$
where β is a fixed hyper-parameter on step size in the outer loop. Table 1 shows all the symbols associated with the proposed algorithm.
**Algorithm 1** Our meta-learning algorithm.**Require**: p(T): distribution over tasks**Parameter**: α, β: step size hyperparameters1:Initialize θ randomly  2:**while** not done **do**3:   Sample a batch of tasks $\tau^{s}∼p\left(T\right)$  4:$\theta_{0}^{\prime }=\theta$  5:**for** 
i=1,2,..,k 
**do**
6:    Compute $\theta_{i}^{\prime }=\theta_{i-1}^{\prime }-\alpha \nabla_{\theta_{i-1}^{\prime }}$$L_{\tau^{s}}$$\left(f_{\theta_{i-1}^{\prime }}\right)$  7:**end for** 8:Sample another batch of tasks $\tau^{q}$∼p(T)  9:Compute $\theta^{\prime }=\theta_{k}^{\prime }-\alpha \nabla_{\theta_{k}^{\prime }}$$L_{\tau^{q}}$$\left(f_{\theta_{k}^{\prime }}\right)$  10:Update θ←θ+β$\left(\theta^{\prime }-\theta \right)$  11:**end while**

### 3.2. Algorithm Analysis

In this subsection, we provide some analysis to better understand why the proposed algorithm works. We first used a Taylor series to approximate the update performed by our algorithm. Then, the effectiveness of our algorithm is shown via the computation of the expected gradient over task and batch sampling.

Suppose we perform two stochastic gradient descent (SGD) steps on $L_{\tau^{s}}$ and one step on $L_{\tau^{q}}$. Let $\phi_{0}$ denote the initial parameter vector. The updated parameter after two steps can be described as
(4)$$\begin{array}{cc} \phi_{0} & =\theta \end{array}$$
(5)$$\begin{array}{cc} \phi_{1} & =\phi_{0}-\alpha L_{\tau^{s}}^{\prime }\left(\phi_{0}\right) \end{array}$$
(6)$$\begin{array}{cc} \phi_{2} & =\phi_{0}-\alpha L_{\tau^{s}}^{\prime }\left(\phi_{0}\right)-\alpha L_{\tau^{q}}^{\prime }\left(\phi_{1}\right) \end{array}$$

The Taylor expansion of $L_{\tau^{q}}^{\prime }\left(\phi_{1}\right)$ can be described as
(7)$$\begin{array}{cc} L_{\tau^{q}}^{\prime }\left(\phi_{1}\right) & =L_{\tau^{q}}^{\prime }\left(\phi_{0}\right)+L_{\tau^{q}}^{\prime \prime }\left(\phi_{0}\right)\left(\phi_{1}-\phi_{0}\right)+O\left(\alpha^{2}\right) \end{array}$$
(8)$$\begin{array}{cc} \; & =L_{\tau^{q}}^{\prime }\left(\phi_{0}\right)-\alpha L_{\tau^{q}}^{\prime \prime }\left(\phi_{0}\right)L_{\tau^{s}}^{\prime }\left(\phi_{0}\right)+O\left(\alpha^{2}\right). \end{array}$$

The gradient of our algorithm after two gradient steps is defined as
(9)$$\begin{array}{cc} g_{ours} & =\left(\phi_{0}-\phi_{2}\right)/\beta =L_{\tau^{s}}^{\prime }\left(\phi_{0}\right)+L_{\tau^{q}}^{\prime }\left(\phi_{1}\right) \end{array}$$
(10)$$\begin{array}{cc} \; & =\alpha /\beta L_{\tau^{s}}^{\prime }\left(\phi_{0}\right)+\alpha /\beta L_{\tau^{q}}^{\prime }\left(\phi_{0}\right)-\alpha^{2}/\beta L_{\tau^{q}}^{\prime \prime }\left(\phi_{0}\right)L_{\tau^{s}}^{\prime }\left(\phi_{0}\right)+O\left(\alpha^{2}\right). \end{array}$$

For any sampled task τ, let $E_{\tau ,\tau^{s}}\left[L_{\tau^{s}}^{\prime }\left(\theta \right)\right]$ and $E_{\tau ,\tau^{q}}\left[L_{\tau^{q}}^{\prime }\left(\theta \right)\right]$ denote the expected losses with $L_{\tau^{s}}$ and $L_{\tau^{s}}$, respectively. The expected losses with $L_{\tau^{q}}$ after the generalization on $\tau^{s}$ is denoted as $E_{\tau ,\tau_{s},\tau_{q}}\left[L_{\tau^{s}}^{\prime }\left(\theta \right)L_{\tau^{q}}^{\prime }\left(\theta \right)\right]$. The expectation of $g_{ours}$ is therefore described as
(11)$$E\left[g_{ours}\right]=E_{\tau ,\tau^{s}}\left[L_{\tau^{s}}^{\prime }\left(\theta \right)\right]+E_{\tau ,\tau^{q}}\left[L_{\tau^{q}}^{\prime }\left(\theta \right)\right]-\alpha \cdot E_{\tau ,\tau_{s},\tau_{q}}\left[L_{\tau^{s}}^{\prime \prime }\left(\theta \right)L_{\tau^{q}}^{\prime }\left(\theta \right)\right]+O\left(\alpha^{2}\right),$$
if the ratio between α and β is a constant. From Equation (Equation 11), we could find out that the expected loss is minimized over tasks; then the higher-order $E_{\tau ,\tau_{s},\tau_{q}}\left[L_{\tau^{s}}^{\prime \prime }\left(\theta \right)L_{\tau^{q}}^{\prime }\left(\theta \right)\right]$ enables fast learning.

## 4. Experiments

In this section, we evaluate the proposed meta-learning algorithm by establishing two medical image segmentation scenarios. We first introduce the dataset for evaluation and the architecture of the baseline model. Then, we discuss the setup for implementation. After that, we compare the proposed algorithm with two existing meta-learning benchmarks: MAML and Reptile. In the final subsection, we study the hyper-parameter.

### 4.1. Dataset and the Baseline Model

We evaluated the proposed algorithm based on a public dataset from the Medical Segmentation Decathlon. This dataset contains ten segmentation tasks, and each task contains diverse scans on a specific tissue type. All the scans have been labeled and verified by an expert human rater, and with his best attempt to mimic the accuracy required for clinical use. We reshaped each scan to 256×256 and simplified the multi-value annotation to a binary segmentation task. Among all the tasks, we randomly selected eight tasks for evaluation owing to computational overheads and memory issues. Six tasks are randomly selected as source tasks, which were the heart from King’s College London, the liver from IRCAD, the prostate from Nijmegen Medical Centre, and the pancreas, spleen, and colon from the Memorial Sloan Kettering Cancer Center. The remaining two tasks, colon and liver, were selected as target tasks. Two medical image segmentation scenarios were established based on these two tasks. For comparison purposes, the scans related to the source tasks were divided into two groups. The source training set of the first group contained 2611 scans of prostate, pancreas, and spleen. The target training set of the first group consisted of 214 scans which were randomly sampled for the task of the colon. The target testing set contained the remaining 1070 scans. The source training set of the second group contained 2877 scans which were of the prostate, heart, and spleen. The target training set of the second group consisted of 191 scans which were randomly sampled for the tasks of liver. The target testing set contained the remaining 18,791 scans.

U-Net was selected as the baseline model, which is illustrated in Figure 4. This model is composed of three partitions, which are the encoder, skip connections, and decoder. The encoder consists of four down-sampling blocks. Each block consists of the repeated application of two 3×3 convolutions, each followed by batch normalization (BN), a rectified linear unit (ReLU), and a 2×2 max-pooling operation with stride 2 for down-sampling. The 2×2 max-pooling operations are replaced with 2×2 transposed convolutions in the decoder. Skip connections concatenate the feature maps before the max-pooling operation in down-sampling blocks with the output of the transposed convolution in up-sampling blocks, which corresponds to the associated depth.

### 4.2. Implementation

We used cross validation to randomly split either the target training set or the source training set into two subsets. One was for training. The other one was for validation. The volume of the training subset was four times that of the validation subset. Each training set was shuffled in each epoch. We utilized data augmentation to reduce the risk of overfitting. The data augmentation included 0∼180 degree random angle flipping, image moving, cross cutting transformation, and image stretching. We applied the cross-entropy function as a loss function. The batch size and the number of epochs were set to 8 and 300, respectively. The batch size in the meta-training phase was set to 6. During meta-training phase, we adopted SGD for each batch with category equipartitioning. The initial learning rate α in algorithm 1 was set as 1 × 10${}^{-3}$. The step size β and the gradient step *k* were set as 0.4 and 3, respectively. We implemented the experiment with Keras. The implementation was performed with an Ubuntu system which employed an NVIDIA GeForce 1080 Ti graphics card which had an 11 Gigabyte memory.

The implementation of the proposed meta-learning algorithm and the two benchmarks relates to pipeline III which is depicted in Figure 5c. In pipeline I, the baseline model is trained directly on the target training set with random initialized parameters. The training phase of pipeline II starts from the parameter vector obtained from the pre-training phase on source domains. Pipeline III applies meta-learning algorithms on source training set.

For MAML, in either the colon or the liver task, we first equally split the source tasks into two partitions where each partition contained all the three classes. The two partitions were used for either support or query purposes. The support and query batches were randomly selected from the support and the query sets, respectively. Each batch contained two examples with three classes. For Reptile, in either the colon or the liver task, each batch was randomly selected from the source tasks, which included two examples with three classes. In our approach, we bring in the layer-freezing technique to optimize the whole feature space. Specifically, when we transfer the initialized parameter vector generated by source domain to target domain, we first train the baseline model on the target training set $T_{tr}^{target}$ with the first two down-sampling blocks which has been frozen. We set the learning rate at this phase as 1 × 10${}^{-3}$. Then, the second block is unfrozen and the target training set $T_{tr}^{target}$ is utilized again. We set the learning rate and the decay rate at this phase as 1 × 10${}^{-3}$ and 0.0077, respectively. For the third time, the target training set $T_{tr}^{target}$ will be applied to the optimization process with entire trainable parameters adjustable. The learning rate at this phase is set as 1 × 10${}^{-4}$.

### 4.3. Experimental Results

Table 2 shows the experimental results of the established two segmentation scenarios. The segmentation performance is described with Dice score (DSC), precision, and recall. DSC is a method which measures the overlap between any two images. DSC has been widely used to evaluate the performance of medical image segmentation when ground truth is available. For binary segmentation of medical images, we set the ratio on target area as 0.9. The ratio on background is 0.1.

DSC can be calculated with the bottom equation
(12)$$DSC=\sum_{k}^{K}\frac{2\omega_{k}\begin{array}{c} \sum_{i}^{N}p_{\left(k,i\right)}g_{\left(k,i\right)} \end{array}}{\begin{array}{c} \sum_{i}^{N}p_{\left(k,i\right)}^{2} \end{array}+\begin{array}{c} \sum_{i}^{N}g_{\left(k,i\right)}^{2} \end{array}},$$
where *N* represents the pixel number. $p_{\left(k,i\right)}\in \left[0,1\right]$ and $g_{\left(k,i\right)}\in \left\{0,1\right\}$ denote the predicted probability and the ground truth label of class *k*, respectively. *K* is the number of class and $\omega_{k}$ denotes the weight of class *k*. The task of semantic segmentation is to predict the class of each pixel in an image. Precision effectively describes the purity of our positive predictions relative to the ground truth. Recall describes the completeness of our positive predictions relative to the ground truth. The bold values in Table 2 indicate that the most suitable freezing depth is 2, since the DSC achieved by this setting is the best.

As shown in Table 2, in the segmentation scenario of colon, we first implemented pipeline I, where the baseline model is trained on the target training set with standard supervised training (SST) without the use of any source training sets. The DSC achieved by this method was 0.537. Then, we implemented pipeline II by respectively pre-training the baseline model on three different source training sets which were the tasks on the prostate, pancreas, and spleen. After the pre-training phase on each task, we trained on the target training set with the SST method. The DSCs of the three tasks were 0.591, 0.590, and 0.591, respectively. We use Δ to represent the DSC improved upon the baseline method. We also implemented pipeline II by pre-training on a batch that contained all the three tissue types; we name this case multi-source I. The DSC achieved by this case was 0.611. The DSCs achieved by MAML, Reptile, and our proposed algorithm were 0.615, 0.608, and 0.628, respectively. We also evaluated the performance of each method when the layer-freezing technique was combined. On average, the layer-freezing technique improved DSC by approximately 4.5% for pipeline II in the cases of multi-source I, MAML, Reptile, and our proposed algorithm. Our proposed algorithm performed the best in terms of DSC, which respectively surpassed pipeline II, MAML, and Reptile by 2.3%, 2%, and 2.4%. The bold values in Table 2 represent the results achieved by our algorithm combined with the layer-freezing technique. The DSC achieved is 0.675, which is an improvement of 0.138 upon the baseline method, and it performed the best among all the methods.

In the segmentation scenario of liver, the DSC achieved by pipeline I was 0.904. The DSCs achieved by pipeline II with single-source pre-training were 0.903, 0.902, and 0.905, respectively. The DSC achieved by pipeline II with the case of multi-source II was 0.905. The DSCs achieved by MAML, Reptile, and our proposed algorithm were 0.904 and 0.912, respectively. On average, the layer-freezing technique improved DSC by approximately 1.2% for pipeline II in the case of multi-source II, MAML, Reptile, and our proposed algorithm. Our proposed algorithm performed the best in terms of DSC, which respectively surpassed pipeline II, MAML, and Reptile by 1%, 1%, and 0.9%. The DSC achieved by our algorithm that used the layer-freezing technique was 0.926, which is 0.022 better than the baseline method.

Figure 6 displays the convergence of loss function on the three approaches under the segmentation scenario of colon. We use red, blue, purple, and green lines to represent our algorithm, pipeline II with the case of multi-source I, MAML + layer-freezing, and Reptile + layer-freezing, respectively. The three approaches all trained on the target training set of colon in the last phase of layer-freezing transfer. We found that all the methods were almost converged in approximately 300 epochs. The computational cost of the proposed framework is introduced by the number of Giga-bytes Floating-point Operations per second (GFLOPs). We estimated the number of GFLOPs and parameters by calling the THOP library function. The number of GFLOPs of the CNN model was 10.112532480. The number of parameters was 4,320,609.

We also calculated the GFLOPs of the CNN model (U-Net) to introduce the computational costs behind the proposed framework. We called the THOP library function to estimate the number of FLOPs and parameters. The GFLOPs of the model was 10.112532480. The number of parameters was 4,320,609.

Figure 7 shows the subjective measures of different approaches in the two established segmentation scenarios. We use green and red contours to represent the image features of ground truth and generated images. For each tissue type, we selected the sample where its segmented image feature was the most visually similar to the ground truth label. From a subjective view, the image produced by our algorithm is visually more similar to the ground truth label and more accurate than the images produced by the other three approaches. For example, in the segmentation task of colon, pipeline II significantly under-estimated the sizes of both the lumen and walls of colon compared to the ground truth labels, while our algorithm reproduced ground truth labels reliably. Even though contour drawing using MAML produced grossly similar results as ground truth, there is an apparent fusion of two adjacent bowel loops. In the segmentation task of the liver, the portal vein (the area circled by an orange circle in raw scans) was mistakenly included as part of the liver parenchyma in (c), (d), and (e), while the ground truth labels in (b) and our algorithm correctly excluded it from liver parenchyma. MAML also generated sharp edges around the gallbladder.

### 4.4. An Ablation Study on the Hyper-Parameter

The initial freezing depth is an important hyper-parameter in layer-freezing, which determines how many layers to be frozen during the training on the target domain. A shallow model will lead to a catastrophic forgetting problem which may destroy the experience learned by meta-learning algorithms. More and more experiences could be obtained and fine-tuned by using a deeper model, but this still cannot guarantee one to achieve a better performance on segmentation, not to mention a higher computational cost. To address this problem, we studied the initial freezing depth under the two few-shot segmentation tasks with an ablation study.

In this study, we investigate the DSCs achieved under three different initial freezing depths. The results are displayed in Table 3. In the case that the depth is one, the learning rates of the first stage and the second stage are 1 × 10${}^{-3}$ and 1 × 10${}^{-4}$, respectively. In the case that the depth is two, the learning rates of the first two stages and the third stage are 1 × 10${}^{-3}$ and 1 × 10${}^{-4}$, respectively. We set the decay rate of the second stage as 0.0077. In the case that the depth is three, the learning rates of the first three stages and the fourth stage are 1 × 10${}^{-3}$ and 1 × 10${}^{-4}$, respectively. We set the decay rate of the third stage as 0.0077. By observing the results shown in Table 3, in the two established segmentation scenarios, we found that the segmentation performance was the best when the depth was set as 2. Although the DSC in the case that the depth was three was better than the case when the depth was one, a deeper depth leads to a higher computational cost.

In [24], the authors point out that using only one gradient descent is not effective during the learning process. The reason is that it optimizes the expected loss over all tasks. It turns out that the performance achieved by the two-step Reptile is worse than the performance achieved by two-step standard supervised learning. However, with more inner loop steps, the performance of Reptile can be further improved and surpass the standard supervised learning. The authors also show that the learning performance is the best when the number of inner loops is four. We therefore set the number of inner loops as four when we implemented Reptile. We set the number of inner loops as three for our algorithm such that the number of gradient descent calculations was the same for each time the parameter was updated.

## 5. Discussion

By observing the results shown in Table 2, we could find out that DSC can be improved when pre-knowledge is used. In the meantime, a better DSC can be achieved when the source training set is sampled from a diverse task distribution. The DSCs achieved by MAML and Reptile are almost same as the result achieved by pipeline II with multi-source training. What is more, the combination of our proposed algorithm and the layer-freezing technique achieved the best performance in the two established scenarios among pipeline I, pipeline II with single-source training, pipeline II with multi-source training, MAML, and Reptile. Specifically, our algorithm achieved 13.8% and 2.4% better DSCs than pipeline I and Reptile, respectively. All our results are reported as averages over five independent runs and with 95% confidence intervals.

In Figure 6, we can see that the losses in the three approaches can be converged to a smaller value in contrast to our algorithm. However, our algorithm achieved a higher DSC than the other three approaches. This observation implies that our algorithm can do a better job of alleviating the overfitting problem for few-shot segmentation tasks.

## 6. Conclusions

This paper proposes a novel meta-learning algorithm to adjust the optimization algorithm so that the segmentation model is nicely learned from a target task which has few examples. Specifically, this algorithm can learn from diverse segmentation tasks across the entire task distribution. In contrast to existing meta-learning algorithms, the proposed algorithm augments the capability to learn from the diversity of image features which characterize a specific tissue type while showing diverse signal intensities. To demonstrate the effectiveness of the proposed algorithm, extensive experiments were conducted by using a diverse set of segmentation tasks on two optimization-based meta-learning benchmarks. The experimental results show that our algorithm surpasses the two benchmarks and brings consistent improvements to both DSC and subjective measures, which implies that the proposed algorithm can produce a better generalization of the target task which has few examples.

## Figures and Tables

**Figure 1 jimaging-07-00031-f001:**
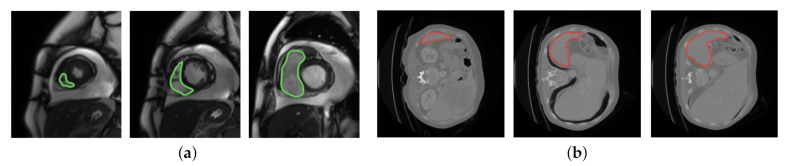
The diversity of image features in two example tissue types: heart (**left**) and spleen (**right**). The image features are segmented by green and red lines in the heart and spleen, respectively. (**a**) Heart; (**b**) spleen.

**Figure 2 jimaging-07-00031-f002:**
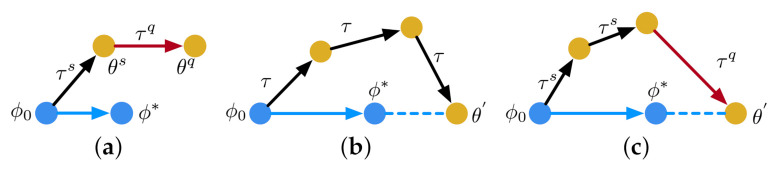
Optimization-based meta-learning algorithms. We use black and red arrows to represent the gradient steps in the inner loop and the outer loop, respectively. The blue arrow represents the direction of model update. (**a**) MAML; (**b**) Reptile; (**c**) our algorithm.

**Figure 3 jimaging-07-00031-f003:**
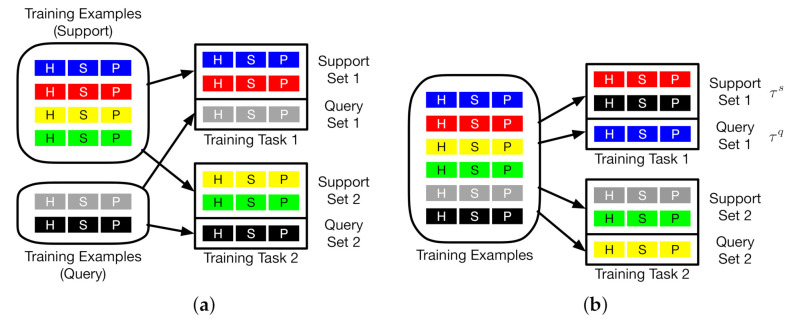
The selection of examples in the meta-training phase. Left: MAML. Right: Our algorithm. H, S, and P represent the MRI images from the heart, spleen, and prostate, respectively. Each example is displayed by a particular color. (**a**) MAML; (**b**) ours.

**Figure 4 jimaging-07-00031-f004:**
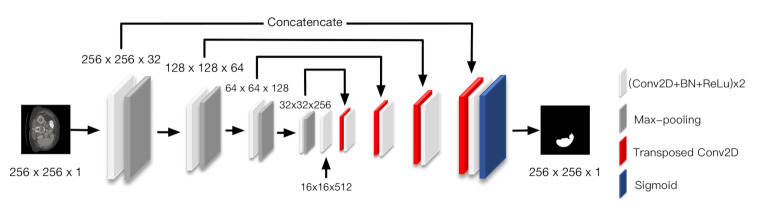
U-Net architecture.

**Figure 5 jimaging-07-00031-f005:**
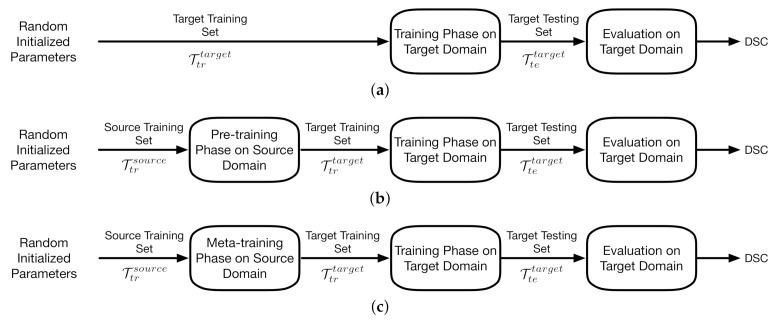
The three pipelines. (**a**) Pipeline I; (**b**) pipeline II; (**c**) pipeline III.

**Figure 6 jimaging-07-00031-f006:**
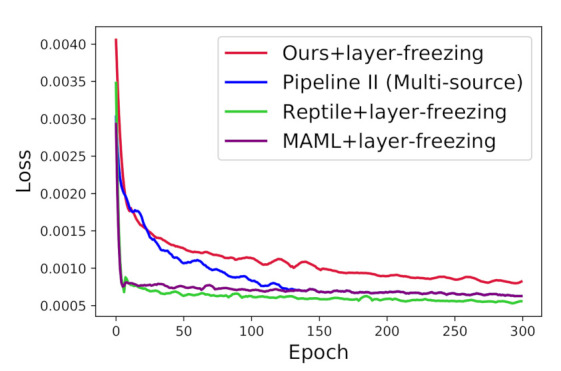
The convergence of three different approaches under the few-shot scenario of colon.

**Figure 7 jimaging-07-00031-f007:**
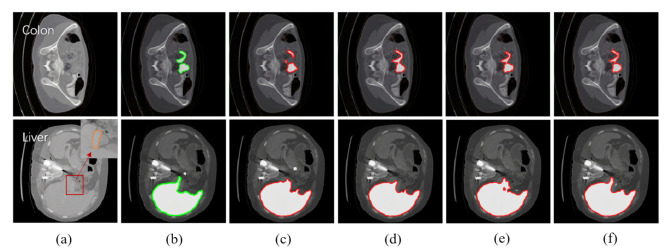
Subjective measures of different approaches in the established segmentation scenarios (top: colon, bottom: liver). Each row presents one typical task, from left to right: (**a**) raw scans; (**b**) ground truth labels; (**c**) pipeline II (multi-source Training); (**d**) MAML + layer-freezing; (**e**) Reptile + layer-freezing; (**f**) ours + layer-freezing. White areas represent the image features of colon and liver.

**Table 1 jimaging-07-00031-t001:** All the symbols associated with the proposed algorithm.

Symbol	Description
θ	The initial parameter vector
$\theta^{\prime }$	Updated parameter vector (out loop)
α	Learning rate (inner loop)
β	Step size (outer loop)
p(T)	The source training set
$\tau^{s}$	Support batch
$\tau^{q}$	Query batch
L	Loss function
*f*	The parametrized function
*∇*	Gradient descent steps

**Table 2 jimaging-07-00031-t002:** The results from two few-shot scenarios.

Source Domain		Target Domain		Dice Coefficient	Δ	Precision	Recall
Task (s)	Method	Task (s)	Method				
Null	Null	Colon	SST	0.537±0.356	-	0.573±0.362	0.573±0.396
Prostate	Pre-training		SST	0.591±0.332	0.054	0.595±0.326	0.655±0.377
Pancreas	Pre-training		SST	0.590±0.347	0.053	0.634±0.338	0.622±0.384
Slpeen	Pre-training		SST	0.591±0.318	0.054	0.594±0.313	0.663±0.362
Multi-source I	Pre-training		SST	0.611±0.325	0.074	0.629±0.318	0.661±0.362
			Layer-freezing	0.652±0.303	0.115	0.653±0.296	0.717±0.335
	MAML		SST	0.615±0.323	0.078	0.655±0.327	0.659±0.342
			Layer-freezing	0.655±0.306	0.118	0.658±0.301	0.716±0.308
	Reptile		SST	0.608±0.336	0.071	0.652±0.332	0.637±0.367
			Layer-freezing	0.651±0.308	0.114	0.652±0.305	0.714±0.341
	Our Algorithm		SST	0.628±0.323	0.091	0.644±0.319	0.673±0.358
			Layer-freezing	0.675±0.292	0.138	0.669±0.288	0.741±0.322
Null	Null	Liver	SST	0.904±0.169	-	0.891±0.165	0.934±0.172
Prostate	Pre-training		SST	0.903±0.176	−0.001	0.902±0.163	0.923±0.184
Heart	Pre-training		SST	0.902±0.176	−0.002	0.894±0.172	0.928±0.175
Slpeen	Pre-training		SST	0.905±0.163	0.001	0.892±0.174	0.935±0.170
Multi-source II	Pre-training		SST	0.905±0.168	0.001	0.888±0.170	0.942±0.157
			Layer-freezing	0.916±0.157	0.012	0.904±0.156	0.944±0.152
	MAML		SST	0.905±0.167	0.001	0.895±0.172	0.936±0.168
			Layer-freezing	0.916±0.158	0.012	0.905±0.154	0.943±0.160
	Reptile		SST	0.904±0.175	0	0.896±0.172	0.927±0.190
			Layer-freezing	0.917±0.158	0.013	0.907±0.156	0.943±0.158
	Our Algorithm		SST	0.912±0.159	0.008	0.896±0.161	0.944±0.150
			Layer-freezing	0.926±0.141	0.022	0.914±0.143	0.952±0.133

**Table 3 jimaging-07-00031-t003:** The results on the two established segmentation scenarios under different initial freezing depths. The bold values represent the most suitable depths that achieve the best DSCs.

DSC	Target Domain	Colon	Liver
Depths			
1		0.660±0.306	0.920±0.160
2		0.675±0.292	0.926±0.141
3		0.671±0.298	0.924±0.148

## Data Availability

The proposed algorithm was tested on the Medical Segmentation Decathlon, see http://medicaldecathlon.com/ (accessed on 8 February 2020).

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
