# Peer review of "Domain Adaptation for Medical Image Segmentation: A Meta-Learning Method"

_2313-433X, 2021, doi:10.3390/jimaging7020031_

Round 1

Reviewer 1 Report

1.The authors have not clearly described why their method enhances the capability of learning.

  1. The author have not compared and discuss advantages and limitations of each algorithm.
  2. In table 1, the authors should describe evaluation metrics (precision, recall, …) and explain bold values.
  3. The author should provide a more clearly picture on their algorithm.
  4. In figure 6, please draw ground truth on binary image and ground truth contours on predicted map.
  5. The implementation section is not clearly described. In section 4.1 Dataset and the Baseline Model, the author should clarify more clearly the dataset instead of just the number of training and testing sets.

  6. There is one paragraph in section 4.3, the author has compared methods based on subjective measures with a few samples (Fig 6).

Author Response

We thank the reviewer for her insightful comments and suggestions. Please see the attachment. All the comments have been individually replied. The revisions in the updated version are in italics.

Reviewer 2 Report

In this paper, a meta-learning algorithm is introduced to augment 10 existing algorithms with the capability to learn from a diverse segmentation tasks across the entire 11 task distribution. The meta-learning benchmarks which include the Model Agnostic and Meta-Learning (MAML) as well as the Reptile are utilized to evaluate the performance of the developed method. 

The research topic is meaningful but there are some concerns need to be addressed in the revision:

1. The proposed algorithm trains the initial parameter vector on a batch of support sets by multiple gradient steps. It seems the effect of this step is the same as the pre-training process. The authors are suggested to provide detailed explanations on this issue. 

2. Please discuss the influence of diverse signal intensities. Why it benefits the meta-learning methods?

3. The mathematical expression should be checked carefully, especially the definition of the symbols. 

4. The performance indicators should be discussed with detailed explanations. In this paper, the DSC is utilized, nevertheless, the reviewer believes that it is not sufficient to prove that the developed methods is better than other methods. Also, the GMACs or GFLOPs metrics should be taken into account. 

Author Response

(The authors gave the same response as above.)

Round 2

Reviewer 2 Report

The reviewer satisfies the response to the comments raised in the first round but still has some concerns about comment 4.

The reviewer believes that the computational cost is very important even in deep learning applications. Hence, the authors must provide convincing evidence and discuss the computational cost of the proposed method. In addition, the GFLOP indicator has been widely used in CNN-based methods for performance evaluation. Please do carry out the experiments.

Author Response

We are grateful for the insightful comment proposed by the reviewer. Please see the attachment. The revised details are formatted as Italic in the updated version. All the comments have been replied.
